# Lightweight SCL-YOLOv8: A High-Performance Model for Transmission Line Foreign Object Detection

**DOI:** 10.3390/s25165147

**Published:** 2025-08-19

**Authors:** Houling Ji, Xishi Chen, Jingpan Bai, Chengjie Gong

**Affiliations:** School of Computer Science, Yangtze University, Jingzhou 434023, China; jhouling@yangtzeu.edu.cn (H.J.); 2023710680@yangtzeu.edu.cn (X.C.); 2022720765@yangtzeu.edu.cn (C.G.)

**Keywords:** object detection, lightweight network, StarNet, CGLU-ConvFormer, LSCDH

## Abstract

Transmission lines are widely distributed in complex environments, making them susceptible to foreign object intrusion, which could lead to serious consequences, i.e., power outages. Currently, foreign object detection on transmission lines is primarily conducted through UAV-based field inspections. However, the captured data must be transmitted back to a central facility for analysis, resulting in low efficiency and the inability to perform real-time, industrial-grade detection. Although recent YOLO series models can be deployed on UAVs for object detection, these models’ substantial computational requirements often exceed the processing capabilities of UAV platforms, limiting their ability to perform real-time inference tasks. In this study, we propose a novel lightweight detection algorithm, SCL-YOLOv8, which is based on the original YOLO model. We introduce StarNet to replace the CSPDarknet53 backbone as the feature extraction network, thereby reducing computational complexity while maintaining high feature extraction efficiency. We design a lightweight module, CGLU-ConvFormer, which enhances multi-scale feature representation and local feature extraction by integrating convolutional operations with gating mechanisms. Furthermore, the detection head of the original YOLO model is improved by introducing shared convolutional layers and group normalization, which helps reduce redundant computations and enhances multi-scale feature fusion. Experimental results demonstrate that the proposed model not only improves the detection accuracy but also significantly reduces the number of model parameters. Specifically, SCL-YOLOv8 achieves a mAP@0.5 of 94.2% while reducing the number of parameters by 56.8%, FLOPS by 45.7%, and model size by 50% compared with YOLOv8n.

## 1. Introduction

With the enhancement of national comprehensive strength, the power grid has become increasingly developed, and electricity now serves as a fundamental pillar for industrial development across various sectors. However, the security challenges associated with power transmission lines have become increasingly severe. Due to their wide distribution, extended lengths, and prolonged exposure to complex and variable natural environments, transmission lines are highly susceptible to interference from various foreign objects, such as kites, plastic bags, bird nests, and advertising banners [1]. Once these foreign objects come into contact with conductors, they can easily trigger electrical discharge, tripping, short circuits, or line disconnections, potentially leading to large-scale power outages with significant economic losses and social consequences [2]. Therefore, regular inspection of transmission lines for foreign object interference is essential to ensure electrical safety.

In the early stages, power departments relied on manual inspection methods for transmission line maintenance [3]. However, these methods exposed workers to significant risks such as falls from height, mechanical injuries, and electrocution, with particularly severe hazards when operating in remote mountainous areas [4]. In recent years, technologies such as drones, infrared imaging [5], radar systems [6], and robotic inspection platforms [7] have emerged as key tools in the power industry. Among these, UAV inspection equipped with high-definition cameras or infrared imaging devices [8] enables rapid, high-altitude inspections of transmission lines, which facilitates the collection of large volumes of image data. However, weather conditions, environmental variability, and limited visibility often affect image quality, making it difficult to ensure consistently high-quality data acquisition [9].

The collected image data are transmitted to the power dispatching center or operation and maintenance personnel for subsequent manual inspection. However, blurred images may result in missed detections or misidentifications by staff. In addition, issues such as inefficiency and subjectivity can easily lead to omissions [10]. Although existing image processing technologies can assist staff in completing inspections and improving detection accuracy, there are still limitations in real-time processing capabilities, making it difficult to respond promptly to complex emergencies [11]. Furthermore, computational demands and resource consumption of current models continue to increase. Therefore, lightweight model design is crucial for achieving efficient operation in resource-constrained environments.

We propose the SCL-YOLO model, which achieves a lightweight design while maintaining high precision and supporting multi-class foreign object detection. By reducing computational complexity and resource consumption, the model improves operational efficiency.

The main contributions of this study are outlined as follows:1.A dataset comprising 9736 images of transmission line foreign objects, captured via UAV aerial photography, is constructed using preprocessing and data augmentation techniques. The dataset includes six object categories: bird nests, balloons, kites, garbage, insulators, and defective insulators.2.Based on the YOLOv8n framework, the model replaces the original CSPDarknet53 backbone with the StarNet network to reduce computational parameters and model overhead.3.The CGLU-ConvFormer module optimizes the structure by combining the convolution operation and gating mechanism at the backbone network. This enhances the model’s ability to capture local and nonlinear features.4.The Lightweight Shared Convolutional Detection Head (LSCDH) is introduced, in which redundant convolutional layers are replaced by shared convolutional layers combined with group normalization. This design reduces model parameters and FLOPs while enhancing multi-scale feature fusion.

## 2. Related Work

Traditional image processing methods typically rely on hand-crafted features such as color, texture, and shape etc., combined with classical machine learning algorithms for object recognition.

Li et al. [12] proposed a method to localize insulators using contour projection. Zhao et al. [13] used orientation angle detection and binary shape prior knowledge (OAD-BSPK) for insulator localization. Tiantian et al. [14] applied threshold segmentation combined with morphological operations for preprocessing. Reddy, M. et al. [15] developed an insulator condition assessment system using discrete orthogonal transforms (DOST) for feature extraction on the MATLAB R2010a platform. Wu et al. [16] extracted the target texture using the Gray-Level Cohomology Matrix (GLCM) and Fast Gray-Level Cohomology Integration Algorithm (GLCIA). Guifeng Zhang et al. [17] combined Ant Colony Optimization (ACO) with Cluster Analysis for insulator image segmentation. Tomaszewski et al. [18] combined specialized signal analysis techniques with machine learning algorithms for power line insulator detection. Foudeh et al. [19] developed a rotationally invariant detection technique to estimate the rotation angle of insulators by an efficient algorithm that uses a sliding window mechanism for extracting local orientation map features.

Traditional methods detect foreign objects on transmission lines by leveraging features such as color and shape. However, these methods are prone to confusion with the surrounding environment and struggle with detecting small targets, making them inadequate for real-time and accurate foreign object detection. The rapid advancement of deep learning has significantly propelled progress in foreign object detection on transmission lines. Deep learning-based object detection algorithms automatically learn image features, thereby greatly enhancing detection accuracy and efficiency. In recent years, researchers have tailored deep learning models to meet the specific requirements of transmission line foreign object recognition, focusing on improving both recognition accuracy and model lightweight design. These include region-based CNN [20] and YOLO series algorithms [21].

Ren et al. [22] trained the Region Proposal Network (RPN) and Faster R-CNN to share convolutional features, thereby improving detection accuracy. Bobo Helian et al. [23] proposed a method that integrates 2D depth maps with the Faster R-CNN framework, utilizing a hybrid loss function combining cross-entropy and root-mean-square error to enhance model accuracy and spatial awareness in complex scenarios. Yinggang et al. [24] developed an enhanced Faster R-CNN model with optimized RPN and RoI Pooling modules, incorporating multi-scale feature fusion to improve small object detection. This is particularly important for transmission line inspections, where foreign objects are typically small and easily obscured by complex backgrounds such as tree branches or birds. Li et al. [25] replaced the original YOLOv3 backbone (Darknet-53) with the lightweight MobileNetV2 and introduced depthwise separable convolutions in the detection head, significantly reducing model parameters. Wang et al. [26] proposed an enhanced YOLO model by incorporating GhostNetV2 in the backbone and integrating SE and CBAM attention mechanisms. Although the model improves accuracy, it compromises computational efficiency and is not suitable for real-time applications. Li, Han et al. developed a detection algorithm based on an improved YOLOv5s architecture, incorporating the Ghost module to reduce model complexity and KL divergence as the loss function to optimize detection performance. Li et al. [27] optimized the YOLOv5 backbone using DSConv and an improved E-ELAN structure to reduce parameter count. However, the model was only evaluated on a self-constructed dataset, limiting its generalizability. Wang et al. [28] proposed a lightweight YOLO variant using Ghost modules to replace traditional convolutions and introduced the SimSPPF structure to reduce computational burden and parameter size. Yu et al. [29] introduced a hyperparameter-optimized YOLOv7 model using a genetic algorithm (GA) and space-depth (SPD) convolution, de-signed for detecting foreign objects in UAV imagery of transmission lines. Peng et al. [30] designed a lightweight YOLO architecture incorporating a Channel Shuffling Dual-Path Aggregation Network (CSDPAN) and Efficient Intersection over Union (EIoU) to reduce computational and parametric complexity.

Structural health monitoring (SHM) and fault diagnosis models in non-destructive testing (NDT) face similar challenges to those in foreign object detection in transmission lines, particularly in real-time anomaly localization and identifying small targets. Keshun et al. [31] proposed a physically constrained quadratic neural network for bearing fault diagnosis under zero-fault sample conditions, where interpretable feature learning is essential for industrial reliability. Similarly, we can adopt their solutions for small target recognition. Fault diagnosis models in NDT have also been applied to CNN and Transformer architectures for foreign object detection.

In non-destructive testing (NDT), edge deployment constraints drive the design of lightweight models. He et al. [32] achieved real-time insulation angle estimation using deep separable convolutions. However, unlike NDT methods that rely on dedicated sensors, our drone-based approach requires robustness against environmental factors such as lighting and weather changes. In addition to NDT-focused methods that identify small target defects through physical constraints, knowledge distillation (KD) provides an alternative approach for model compression. Yang et al. (2024) [33] recently proposed a student-centric knowledge distillation method that dynamically adjusts the learning focus between the teacher and student models.

Although KD can reduce model size, our method prioritizes structural optimization over post-training compression, as unmanned aerial vehicle (UAV) edge devices lack the resources to host teacher models during inference, and transmitting line objects (such as deformed kites) require structural adaptation beyond feature mimicry. Therefore, we prioritize structural optimization, which eliminates the need for pre-training teacher models and maintains end-to-end trainability.

However, excessive pursuit of model compression resulted in a noticeable drop in detection accuracy. Unlike previous studies, which primarily focus on optimizing the backbone network or introducing attention mechanisms (typically targeting a single module to achieve model lightweighting or detection accuracy improvement), the method proposed in this work conducts systematic optimization across multiple key components. Specifically, while replacing the backbone network, the feature fusion structure and detection head are simultaneously redesigned and optimized. This comprehensive approach achieves a better balance between detection accuracy and computational efficiency under resource-constrained conditions, thereby enhancing the intelligence and safety of multi-class foreign object detection on transmission lines.

## 3. Methods

The YOLO family of algorithms has garnered significant attention in the target detection community. YOLOv8, developed by the Ultralytics team, significantly improves detection accuracy while maintaining highspeed inference performance. Compared to previous versions, YOLOv8 adopts CSPDarknet53 as the backbone and introduces a cross-stage partial connection (CSP) structure. Additionally, by leveraging the feature pyramid network (FPN), it achieves effective multi-scale feature fusion, enhancing its capability to detect targets of various sizes. The introduction of adaptive anchor box computation further improves detection precision. Furthermore, YOLOv8 employs Complete Intersection over Union (CIoU) loss, which more effectively measures the discrepancy between predicted and ground truth bounding boxes, thereby enhancing localization accuracy. Accordingly, YOLOv8n is selected as the baseline model in this study.

Although YOLOv8n demonstrates high accuracy in detecting general objects, the complex backgrounds of transmission line environments, such as sky, vegetation, and buildings, etc., introduce substantial interference, often leading to false detections and missed targets. Moreover, the computational intensity of YOLOv8n limits its inference speed on mobile platforms, making it difficult to meet real-time requirements.

### 3.1. Overall Network Architecture

This work proposes an improved model of YOLOv8n, SCL-YOLOv8, whose architecture is shown in Figure 1.

First, the backbone network of YOLOv8n is replaced with the StarNet architecture, where the Star Block serves as a key structural component. This substitution effectively achieves backbone light weight while maintaining high detection accuracy. Subsequently, CGLU-ConvFormer, a channel mixing module that integrates convolutional operations and gating mechanisms, is introduced to replace the original C2f module in the neck of YOLOv8n, enhancing feature selectivity, nonlinear representation capability, and computational efficiency. Compared with the original C2f module, CGLU-ConvFormer is more lightweight and offers improved performance in multi-scale feature fusion and information processing. Finally, the original detection head of YOLOv8n is replaced with LSCDH, a lightweight detection head. This head incorporates shared convolutional layers and a lightweight design, which not only reduces computational cost but also efficiently processes feature maps across multiple layers. This improvement facilitates better fusion of high-level and low-level feature information, thereby enhancing the model’s multi-scale feature learning capability.

Although previous lightweight YOLO variants, including those incorporating GhostNet, achieved model compression by replacing standard convolutions with lightweight backbones or Ghost modules, they often compromised local detail representation or relied excessively on backbone simplification. In contrast, our CGLU-ConvFormer integrates deep convolutions with gating mechanisms, enhancing spatial feature selectivity—particularly for small objects—without significantly increasing model complexity. Meanwhile, the LSCDH module reduces redundancy in multi-scale detection heads by sharing parameters across scales and incorporating group normalization, a technique rarely used in YOLO heads. These innovations strike a more balanced trade-off between accuracy and efficiency, particularly in cluttered aerial detection environments. YOLOv8n has been designed with a focus on innovation rather than comprehensiveness, aiming to enhance accuracy, efficiency, and deployability in edge scenarios, such as drone-based detection.

### 3.2. StarNet Feature Extraction Network

Traditional YOLO-based target detection networks typically contain deep architectures with a large number of parameters, which adversely affect computational efficiency. Therefore, this study employs the StarNet [34] architecture as the backbone network to reduce computational complexity. StarNet adopts a fourth-order hierarchical structure to extract low-level (e.g., edges, textures), mid-level (e.g., local shapes, simple structures), high-level (e.g., complex structures, semantic information), and global features (e.g., contextual cues, overall object characteristics). Efficient feature extraction is achieved through downsampling operations in the convolutional layers, combined with a modified demo block.

The core operation of StarNet, the star operation, projects features into a high-dimensional implicit feature space, and performs feature fusion across different subspaces via element-wise multiplication. The formulation of the star operation in a single-layer network is as follows:(1)O=(ω1TX) ∗ (ω2TX)ω=ωB.X=X1
where *O* is the result after performing a star operation; *ω* is the weight matrix for the linear layer; *B* is the bias of the linear layer; X is the input, and * represents the star operation.

From this we can define $\omega_{1},\omega_{2}$,$x\in R^{(d+1)*1}$, where *d* is the number of input channels. The star operation is further rewritten as(2)$$(\omega_{1}^{T}X)\;*\;(\omega_{2}^{T}X).$$(3)$$=\left(\sum_{m=1}^{d+1}\omega_{1}^{m}x^{m}\right)\;*\;\left(\sum_{n=1}^{d+1}\omega_{1}^{m}{\omega_{2}^{n}x}^{n}\right).$$(4)$$=\sum_{m=1}^{d+1}\sum_{n=1}^{d+1}\omega_{1}^{m}{\omega_{2}^{n}x^{m}x}^{n}\;\;.$$
(5)$$=\underset{\frac{\left(d+1)(d+1\right)}{2}}{\underbrace{\alpha_{\left(1,1\right)}x^{1}x^{1}+\cdots +\alpha_{\left(4,5\right)}x^{4}x^{5}+\cdots }}+\alpha_{\left(d+1,d+1\right)}x^{d+1}x^{d+1}$$where *m*, *n* are channel subscripts and *α* is a coefficient defined as follows:(6)$$\left\{\begin{array}{c} \omega_{1}^{m}\omega_{2}^{n},i=j\\ \omega_{1}^{m}\omega_{2}^{n}+\omega_{1}^{n}\omega_{2}^{m},i\neq j \end{array}\right.\;\;\;\;\;.$$

Equation (5) consists of $\frac{\left(d+1)(d+1\right)}{2}$ distinct terms and is therefore efficiently computed in d-dimensional space, yet can realize a representation in the implicit feature space of higher dimensions $\frac{\left(d+1)(d+1\right)}{2}\approx {\left(\frac{d}{\sqrt{2}}\right)}^{2}$. This design significantly extends the feature dimensions in a single-layer network without adding computational burden. By stacking multiple layers, the implicit dimensionality can recursively grow exponentially, thus approaching infinite dimensionality. After k layers of star operations, the network can implicitly generate a ${\left(\frac{d}{\sqrt{2}}\right)}^{2^{K}}$ dimensional feature space.(7)$${\;\;\;\;\;\;\;\;\;\;\;\;\;O}_{K}=\omega_{K,1}^{T}O_{K-1}\;*\;\omega_{K,2}^{T}O_{K-1}\in R^{{\left(\frac{C}{\sqrt{2}}\right)}^{2^{K}}}.$$
where $O^{K}$ is the output after the kth layer star operation.

The feature extraction module of StarNet achieves its functionality through a cascade of multiple Star Blocks, as illustrated in Figure 2.

Each Star Block incorporates a dual-branch linear transformation structure, and the outputs from both branches are fused through element-wise multiplication to form a nonlinear feature representation. Residual connections are introduced to enhance training stability and alleviate the vanishing gradient problem common in deep neural networks. A progressive downsampling strategy is adopted in the layer design, where the number of channels is increased via convolution while the spatial resolution is halved at each stage. This design enlarges the receptive field and improves the model’s multi-scale feature representation capability, as shown in Figure 3.

In addition, the normalization strategy is optimized by using batch normalization in place of layer normalization, which effectively normalizes feature distributions after deep convolution, alleviates internal covariate shift, improves training stability, and suppresses overfitting. Furthermore, the integration of depthwise separable convolutions at the end of each module preserves feature extraction capability while significantly reducing computational cost and parameter count. Therefore, the StarNet architecture is selected as the backbone network in this study to maintain high-performance inference while reducing model complexity.

### 3.3. CGLU-ConvFormer

C2f is a cross-stage feature fusion module based on convolution and residual joining, which is mainly used for feature fusion and multi-scale feature extraction, but its feature extraction capability is limited when dealing with small targets and complex backgrounds. In addition, the computational complexity of the C2f module is high, especially when dealing with high-resolution images, and the inference speed will be significantly slower. In this paper, CGLU-ConvFormer is designed to replace the C2f module. Its structure is shown in Figure 4.

ConvFormer [35] uses MetaFormer as the base architecture and adopts a 4-stage hierarchical architecture, where features are processed in each stage by downsampling and a different number of modules, which ultimately realizes an efficient image feature extraction and classification process. First, the input data is transformed into a sequence of features by input embedding operation. The embedding process is represented as:(8)$$X=InputEmbedding\left(I\right)$$

The obtained feature sequences are input into repetitive ConvFormer blocks, the structure of which is shown in Figure 5. In each ConvFormer block, a normalized Norm (*X*) operation is first performed, followed by the use of a token mixer based on depth-separable convolution, i.e., according to the following equation.(9)$$Convolutions\left(X\right)=p\omega_{2}(d\omega (\sigma (p\omega_{1}(X))))$$
where $p\omega_{1}$ denotes the first point-by-point convolution of the input feature *X*, $p\omega_{2}$ denotes the second point-by-point convolution of the input feature *X*, dω denotes the deep convolution, and σ denotes the nonlinear activation function applied after the convolution operation.

In the ConvFormer architecture, the Token Mixer serves as a core component that facilitates the interaction and fusion of features from different locations. The result obtained by the Token Mixer is then summed with the residuals and normalized for the second time, multiplied with the learnable parameters and then multiplied with the nonlinear activation function, and then multiplied with the learnable parameters, and the result is then summed with the residuals; after that, it undergoes a 4-stage process, each of which consists of more than one of the above mentioned ConvFormer blocks, and downsampling is performed by the downsampling module, the first of which is a convolutional kernel of size 7 with a step size of 4. The first downsampling module has a convolutional kernel size of 7 and a step size of 4, while the last three downsampling modules have a convolutional kernel size of 3 and a step size of 2. Finally, the features processed in the four stages are passed through the classifier header to obtain the final output.

In the ConvFormer module, the Channel MLP is employed to facilitate information transfer and feature transformation along the channel dimension. Channel dependencies are captured through a two-layer fully connected network (MLP) that performs nonlinear transformations. However, the Channel MLP exhibits several limitations: it cannot capture local spatial information, it incurs high computational complexity, and possesses a rigid structure that limits dynamic adaptability. To address these issues, this paper proposes replacing the Channel MLP in the ConvFormer module with a Convolutional Gated Linear Unit (CGLU) to enhance both performance and efficiency.

CGLU functions as a channel mixer by integrating convolutional operations with gating mechanisms, thereby enhancing local modeling capability and robustness. The Gated Linear Unit (GLU) [36] is a channel mixing mechanism composed of two linear projections. CGLU extends GLU by incorporating a 3 × 3 depthwise separable convolution before the activation function in the gated branch. This operation captures the local features of each token’s nearest neighbors, thereby providing CGLU with a significant advantage in modeling local details, as formulated below:(10)$$CGLU\left(X\right)=\left(X\omega_{1}+B_{1}\right)⨀GELU\left(DWC\left(X\omega_{2}+B_{2}\right)\right)$$
where *B* is the bias, ⨀ is element-by-element multiplication, *DWC* is depth-separable convolution, and GELU is the activation function.

As illustrated in Figure 5, CGLU processes the input through an initial linear transformation layer. In the gating branch, the input first undergoes a 3 × 3 depthwise convolution, which captures local information at each position of the input feature map and generates a unique gating signal based on the nearest-neighbor features of each token. The output is then passed through a GELU activation function to obtain the final gating signal. The original input and the gating signal are combined through element-wise multiplication, allowing the gating signal to regulate the flow of information. This enables the model to adaptively modulate the input based on local features, thereby generating more representative outputs.

### 3.4. LSCDH

The YOLOv8n detection head adopts a decoupled architecture, processing the three-scale feature maps from the backbone and neck by dividing them into a classification branch (Cls.) for class probability prediction and a regression branch (Bbox.) for object localization. This design, when applied to foreign object recognition on transmission lines, involves three separate detection heads for multi-scale prediction, which introduces redundancy in output parameters. To address this issue, we propose a redesigned Lightweight Shared Convolutional Detection Head (LSCDH), specifically optimized for the target detection task. The goal is to reduce computational cost while maintaining high performance by employing shared convolutional layers and a lightweight structure. The overall architecture is illustrated in Figure 6.

The most prominent feature of the LSCDH detection head is the shared convolutional layer, which integrates the feature extraction processes of multiple tasks into a unified operation. This design reduces computational redundancy and simplifies the overall model structure. In LSCDH, a shared convolutional kernel is used to integrate the convolution operation with Group Normalization (GN) [37], forming a unified module termed Conv Group Normalization (Conv-GN). First, the input feature maps are processed by a Conv2d layer to extract and generate enhanced feature representations. Subsequently, group normalization is applied to the convolutional output by dividing the channels into groups and normalizing each group such that its mean is zero and standard deviation is one. The normalization formula is as follows:(11)$$\hat{x_{i}}=\frac{x_{i}-u_{i}}{\sqrt{\sigma_{i}^{2}+\epsilon }}\cdot \gamma +\beta \cdot$$
where $x_{i}$ are the input features, $u_{i}$ and $\sigma_{i}$ are the mean and variance of each set of features, γ and β are the learnable parameters, and ϵ is a small constant for numerical stability.

Both convolutional layers and group normalization are inherently linear operations. Stacking multiple linear layers results in an overall linear transformation, which limits the model’s ability to capture complex patterns and nonlinear features. To address this limitation, we introduce the Sigmoid-Weighted Linear Unit (SiLU), a nonlinear activation function that enhances the model’s ability to capture complex representations.

The LSCDH receives multi-resolution feature maps (P3, P4, and P5) from the backbone and neck, and applies channel adjustment and feature fusion using a 1 × 1 Conv_GN operation, which facilitates information interaction across channels. Subsequently, Conv_GN with two shared 3 × 3 convolutional kernels is used for feature aggregation, which effectively reduces redundant information and enhances the learning of local spatial features. The shared convolutional layer preserves both the fine-grained gradients from the high-resolution feature map (P3) and the contextual gradients from the low-resolution map (P5), thereby enhancing sensitivity to small targets. In traditional decoupled detection heads, each feature scale requires an independent convolutional layer for processing. The total number of parameters is calculated as follows:(12)$$Params=3\times C_{in}\times C_{out}\times K\times K$$

All the feature scales P3, P4, and P5 in LSCDH share the same set of convolutional kernels for feature extraction, and the size of their total number of parameters is:(13)$$Params=C_{in}\times C_{out}\times K\times K\;$$
where $C_{in}$ is the input feature dimension, $C_{out}$ is the output feature dimension, and the convolution kernel size is *K* × *K*.

It can be noticed by Equations (12) and (13) that the number of parameters is reduced by 1/3 with the use of shared convolutional layers.

Finally, the features extracted from the shared convolution are fed into Conv_Reg and Conv_Cls for regression and classification, respectively, predicting the target’s bounding box coordinates and class probabilities. The Scale layer further enhances the representation and preservation of multi-scale features, enabling the model to adapt to variations in feature distributions, improve localization and classification accuracy, and maintain training stability under various data augmentation strategies.

## 4. Experiments and Analysis

### 4.1. Experimental Environment

The experiments in this paper were conducted using Python 3.8 and PyTorch 1.11.0. The experimental environment includes a 12-core Intel (R) Xeon (R) Platinum 8352V CPU @ 2.10 GHz, 90 GB of RAM, a GPU with 32 GB of memory, and CUDA version 11.3. The Stochastic Gradient Descent (SGD) algorithm was used to optimize the model. The specific training hyperparameters used in this study are summarized in Table 1.

### 4.2. Dataset

Since no publicly available dataset specifically targets foreign objects on transmission lines, the dataset used in this study comprises two components. The first part includes 848 images from the publicly available Chinese Power Line Insulator Dataset (CPLID) [38], containing 227 normal and 621 defective insulators; defective insulators refer to damaged or missing insulators on power transmission lines. The second part was collected through drone-based aerial photography and online sources, focusing on common foreign objects found on transmission lines, including bird nests, balloons, kites, garbage, insulators, and defective insulators, to enrich the dataset and ensure diversity. The collected images exhibit wide variability, with multiple foreign objects often appearing in a single image. Each image was manually annotated with one or more of the following categories: bird nests, balloons, kites, garbage, insulators, and defective insulators, as illustrated in Figure 7.

The dataset covers various conditions, including varying weather, background environments, and altitudes. This diversity enhances the generalization capability of the YOLOv8 model in real-world scenarios. To further improve model generalization, data augmentation techniques were applied during preprocessing. Random rotation, cropping, and scaling were used to increase data diversity. Noise injection was also employed to simulate visibility changes under different weather conditions and occlusions, improving the model’s robustness, as shown in Figure 8.

After preprocessing, the dataset comprises 9736 images, each with a high-resolution size of 8688 × 5792 pixels. A total of 16,532 object annotations are included in the dataset. Annotation was performed using LabelImg1.8.6 software, labeling six types of foreign objects and saving the results in XML format. Specifically, the dataset includes 5934 labels for bird nests, 1742 for balloons, 527 for kites, 595 for garbage, 6581 for insulators, and 1227 for defects, as illustrated in Figure 9. The dataset was randomly split into training, validation, and test sets in an 8:1:1 ratio for model training.

The left panel of Figure 10 shows a data distribution heatmap, where most data points are concentrated in the central region, and the darker color indicates higher point density. This suggests that object centers predominantly cluster near the middle of the image. The right panel displays a scatter plot, where the darker color in the lower-left corner reflects a predominance of small-sized objects in the dataset. This phenomenon may be attributed to the drone’s hovering mode during data collection. To mitigate the data imbalance issue, we employed random rotation, cropping, and scaling techniques to diversify the target locations during dataset preprocessing. In the proposed LSCDH module, multi-scale feature fusion was implemented to enhance sensitivity to small targets.

The RailFOD23 [39] dataset is a synthetic collection specifically designed to for the detection of foreign objects on railway power lines. It is based on approximately 400 real images and synthesized using Adobe Photoshop (PS) software to create 412 abnormal power line images, alongside images automatically generated via ChatGPT3.0, AIGC, and data based on Railsem19. The dataset consists of 14,615 images with annotations for four types of foreign objects: plastic bags, floating objects, bird nests, and balloons, amounting to a total of 40,541 labels. As shown in Table 2, compared to the transmission line dataset presented in this paper, the RailFOD23 dataset includes more images. Still, it lacks annotations for defective insulators and has a lower proportion of small targets.

### 4.3. Evaluation Indicators

To comprehensively and specifically evaluate the improvements proposed in this paper for lightweight foreign object detection on transmission lines, precision (*P*), recall (*R*), and mean average precision (*mAP*) at an IoU threshold of 0.5 are adopted. *mAP* evaluates the model’s performance when the Intersection over Union (*IoU*) between the predicted and ground-truth bounding boxes exceeds 0.5, a threshold that is widely used for its generalizability. These metrics provide an intuitive and fair evaluation of detection accuracy. Precision, recall, average precision (*AP*), and *mAP* are employed to comprehensively assess the model’s accuracy and ensure its reliability in recognizing foreign objects. The corresponding formulas are as follows:(14)$$P=\frac{TP}{TP+FP}\cdot$$(15)$$\;R=\frac{TP}{TP+FN}\cdot$$(16)$$\;AP=\int_{0}^{1}P(R)dR\cdot$$(17)$$mAP=\frac{1}{n}\sum_{i=1}^{N}AP_{i}\;\cdot$$
where *TP* is the number of targets in the image that are detected and correct; *FP* is the number of targets in the image that are detected but incorrect; *FN* is the number of targets in the image that are not detected; *n* is the number of target categories in the image.

FLOPS and the number of params are used to evaluate the lightness of the model for visualization. FLOPS represent the number of floating-point operations performed by the computational model in a forward propagation process, which measures the computational complexity. The lower the FLOPS, the lower the computational overhead of the model, which is represented by the following formula:(18)$$FLOPs=2\times H\times W\left(C_{in}K^{2}+1\right)C_{out}$$

*Params* are used to represent the total number of all trainable parameters in the model: the smaller the number of parameters, the lower the storage requirement of the model, which is suitable for mobile or embedded device deployment; the calculation formula is expressed as follows.(19)$$\;Params=C_{in}{\times K}^{2}\times C_{out\;\;\;}$$

### 4.4. Ablation Experiment

Different improvement strategies vary in their impact on the final model performance. Some optimizations significantly enhance detection accuracy but introduce additional computational overhead, while others prioritize faster inference or improved generalization. To comprehensively evaluate the individual contributions of each module to overall model performance, a series of ablation experiments were conducted.

YOLOv8n is used as the baseline model, with its key components sequentially replaced by the StarNet module, the C2f-CGLU-ConvFormer module, and the LSCDH module, to analyze their respective impacts on detection accuracy, computational efficiency, and convergence speed. As shown in Figure 11, experiment A represents the original YOLOv8n, B incorporates the StarNet module, C integrates the C2f-CGLU-ConvFormer module, and D includes the LSCDH module. Experiment E combines StarNet and C2f-CGLU-ConvFormer, F combines StarNet and LSCDH, and G combines C2f-CGLU-ConvFormer with LSCDH. Experiment H represents the model with all three modules integrated. By comparing the results of these ablation experiments, the strengths and limitations of different improvement strategies can be clarified, their contributions to the object detection task quantified, and insights gained to guide further optimization of the network architecture.

The experimental results presented in Table 3 demonstrate that the proposed improvement strategy enhances detection performance to varying degrees. The StarNet module serves as the backbone network, expanding the receptive field and enabling hierarchical abstraction, while significantly reducing both the number of parameters and computational cost. Specifically, the FLOP count is reduced to 27% and 19.8% of the baseline model, respectively. Replacing the baseline C2f module with CGLU-ConvFormer enhances control over the information flow through a gating mechanism, improving the fusion of spatial and channel features. This modification makes the model more effective at handling small objects and complex backgrounds.

Given that the dataset in this study contains a substantial proportion of small targets, incorporating the LSCDH module into the baseline model facilitates the fusion of multi-scale feature maps. This integration results in a 0.3% increase in mAP@0.5. The LSCDH module eliminates redundant convolutional operations, reduces computational overhead, and decreases the number of parameters in the detection head. These optimizations reduce parameters and FLOPs to 23.6% and 19.8% of the baseline, respectively, while the overall model size is reduced by 4 MB, achieving a peak inference speed of 171.9 FPS.

The combination of StarNet and CGLU-ConvFormer (referred to as YOLOv8n-SC in Table 3) improves feature extraction through StarNet hierarchical architecture, which extracts rich multi-scale features and implicitly generates high-dimensional representations. These features are then passed to the neck stage of CGLU-ConvFormer, which enhances local detail representation and nonlinear feature transformation using gated convolutions and channel mixing. As a result, the model’s accuracy improves, with mAP@0.5 increasing to 94.6%. The LSCDH module, located in the detection head, utilizes shared convolutions and group normalization to decode multi-resolution features more efficiently, while minimizing redundancy. The sequential deployment of these modules ensures a progressive improvement in feature quality from input to output.

Compared to the original YOLOv8n, the proposed model achieves 94.2% mAP@0.5, with parameters, FLOPs, and model size reduced to 56.8%, 45.7%, and 50% of the baseline, respectively. These results from ablation studies confirm the benefit of multi-module collaboration in the proposed architecture. The enhanced model demonstrates a more accurate focus on target regions, especially for small and occluded objects.

The combined effects of Enhanced Hierarchical Extraction (StarNet), Local Refinement (CGLU-ConvFormer), and Multi-Scale Decoding (LSCDH) enable the network to better distinguish true targets from background noise. The proposed model not only ensures high detection accuracy for foreign objects on power lines but also exhibits exceptional lightweight characteristics, making it highly suitable for deployment on resource-constrained edge devices. This allows for real-time performance while reducing model complexity and improving computational efficiency.

### 4.5. Generalization Evaluation

To assess the broader applicability of SCL-YOLOv8 beyond power line detection, we performed a cross-domain evaluation using the VisDrone benchmark (The Vision Meets Drone Dataset) [40]. This dataset, released by the Tianjin University team, comprises 10,209 high-resolution static images captured from aerial perspectives using drones. It encompasses ten categories: ‘pedestrian’, ‘people’, ‘bicycle’, ‘car’, ‘van’, ‘truck’, ‘tricycle’, ‘awning-tricycle’, ‘bus’, and ‘motorcycle’. The dataset encompasses a range of diverse environments, including urban, suburban, campus, and retail settings. Featuring a high proportion of small objects, it presents several challenges (e.g., strong occlusions, dense object distribution, and significant angle variations), making it an ideal benchmark for evaluating model robustness.

As shown in Table 4, the accuracy and model size of our model on the VisDrone dataset have improved relative to YOLOv8n, with a mAP@0.5 of 47.8%, representing a 13.5% increase over the baseline model. Both the number of parameters and computational complexity have been reduced by over 50%, with the model size compressed to 6MB. A comparative visualization of our model’s detection capabilities in four dataset scenarios is provided in Figure 12. These results demonstrate that our improved model offers enhanced detection capabilities, particularly in terms of target localization and classification accuracy, and is better suited for deployment on resource-constrained edge devices.

### 4.6. Comparison Experimen

To verify the effectiveness of the proposed model, a series of comparative experiments were conducted using several mainstream lightweight object detection models, including YOLOv3-Tiny, YOLOv4-Tiny, YOLOv5n, YOLOv6n, and YOLOv7-Tiny. All models were trained and evaluated on the same dataset to ensure the fairness and consistency of the comparison. The experiments were implemented using the PyTorch deep learning framework under identical hyperparameter settings, including learning rate, number of training epochs, and data augmentation strategies.

As shown in Table 5 and Figure 13, among mainstream lightweight models, including PP-YOLOE, YOLOv3-Tiny, YOLOv4-Tiny, YOLOv5n, YOLOv6n, and YOLOv7-Tiny, SCL-YOLOv8 achieves the highest mAP@0.5 of 94.2%, highlighting its superior detection performance. Detection accuracy progressively increases from 83.2% for PP-YOLOE to 93.7% for YOLOv8n, indicating that continuous improvements in model architecture and training strategies lead to significant performance gains. From the perspective of model light weight, SCL-YOLOv8 also exhibits excellent performance, with only 1.4M parameters, 4.4G FLOPs, and a model size of 3.1MB, substantially lower than those of other models. Compared with YOLOv7-Tiny, the parameters, FLOPs, and model size are reduced by 77.4%, 66.2%, and 74.8%, respectively. Compared with YOLOv5n, these metrics are reduced by 80.6%, 73.3%, and 68.7%, respectively. These reductions enable real-time operation on embedded or edge devices, making the model well-suited for real-world deployment in transmission line scenarios. SCL-YOLOv8 achieves an inference speed of 123.8 FPS, which is slightly lower than YOLOv8n’s 150.6 FPS. However, comparative experiments demonstrate that SCL-YOLOv8 offers a clear advantage under resource-constrained conditions by achieving lightweight efficiency without compromising detection accuracy.

### 4.7. Visualization Analysis

As shown in Figure 14, the top row illustrates the detection results of the baseline YOLOv8n model, while the bottom row displays the results from the model enhanced with both the C2f-CGLU-ConvFormer and LSCDH modules. The comparison demonstrates the enhanced recognition capability of the proposed model. When applied to the same input image, the improved model not only exhibits higher confidence scores compared to the baseline model but also successfully identifies defect targets, such as insulator faults, which are missed by the baseline. This highlights the effectiveness of the combined modules in improving detection accuracy, especially for subtle or small-scale foreign objects.

As shown in Figure 15, the attention regions generated by the baseline YOLOv8n model are relatively dispersed, with unclear target boundaries and unintended activation in background areas. This indicates the model’s insufficient feature extraction capability, particularly when dealing with small objects or cluttered backgrounds. In contrast, the improved SCL-YOLOv8 model demonstrates a stronger ability to accurately focus on key target regions across diverse scenes. It effectively suppresses background interference and exhibits enhanced robustness and localization accuracy, especially in detecting foreign objects and insulators.

Comparing the inference results of YOLOv3-Tiny, YOLOv5n, YOLOv6n, YOLOv8n, and SCL-YOLOv8, the performance of SCL-YOLOv8 is superior in terms of accuracy for transmission line foreign object detection, as shown in Figure 16.

## 5. Conclusions and Future Work

In this paper, we propose SCL-YOLOv8, a lightweight power line foreign object detection model based on an optimized YOLOv8n framework. This model incorporates three key innovations—StarNet, which replaces CSPDarknet53 with star operations for more efficient feature extraction, CGLU-ConvFormer, which enhances sensitivity to small objects through gated convolutions, and LSCDH, which achieves multi-scale fusion via shared convolutions and group normalization. These advancements allow the model to maintain a mAP@0.5 of 94.2% while simultaneously reducing the number of parameters, FLOPs, and model size by 56.8%, 45.7%, and 50%, respectively. These improvements enable real-time deployment on drones (123.8 FPS) without sacrificing accuracy.

In future work, we will focus on optimizing small object detection by designing more efficient multi-scale feature fusion strategies to address the issue of missed detections for small targets in high-resolution images captured by UAVs. We will enhance the model’s focus capability on small objects and employ relevant techniques to generate training data under extreme scenarios (e.g., dense occlusions, harsh weather conditions), thereby improving the model’s generalizability. These improvements are expected further to expand the applicability and value of the proposed method. Such efforts represent critical steps toward achieving fully automated foreign object detection in power transmission lines.

## Figures and Tables

**Figure 1 sensors-25-05147-f001:**
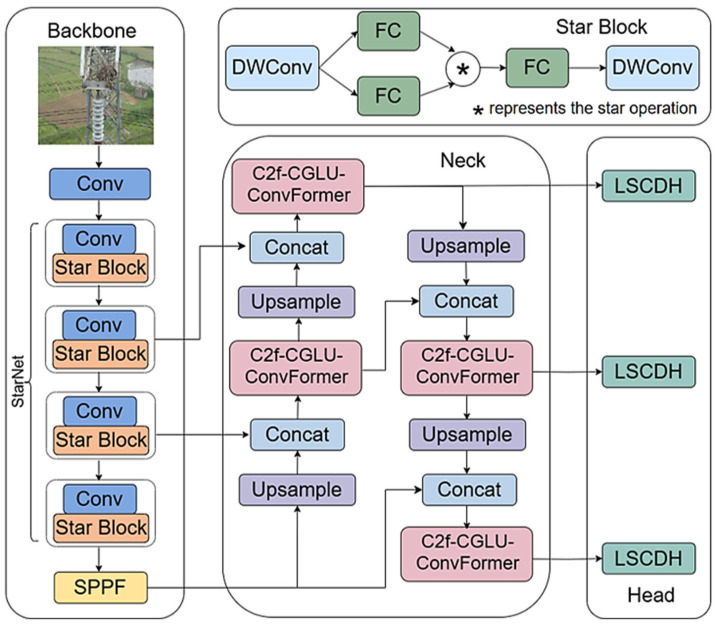
Network architecture of SCL-YOLOv8.

**Figure 2 sensors-25-05147-f002:**
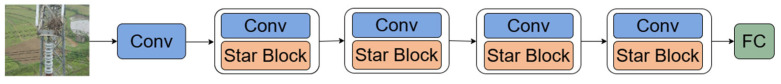
StarNet network structure diagram.

**Figure 3 sensors-25-05147-f003:**
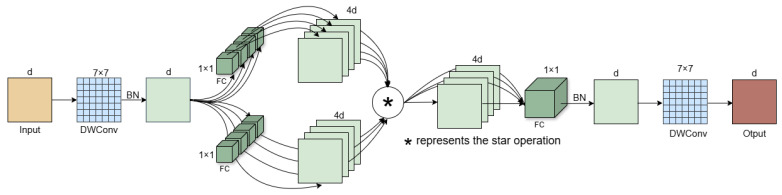
Structure of Star Block.

**Figure 4 sensors-25-05147-f004:**
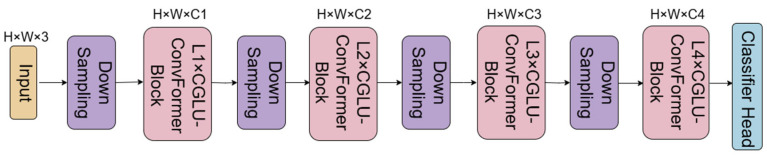
Structure of CGLU-ConvFormer.

**Figure 5 sensors-25-05147-f005:**
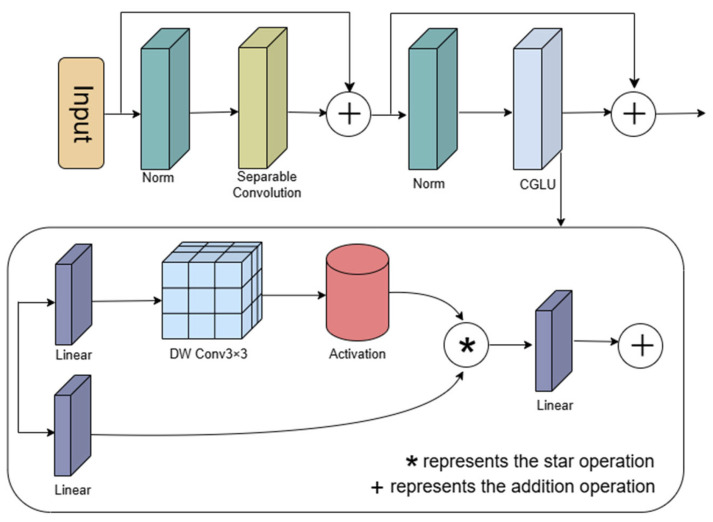
Structure of CGLU-ConvFormer block.

**Figure 6 sensors-25-05147-f006:**
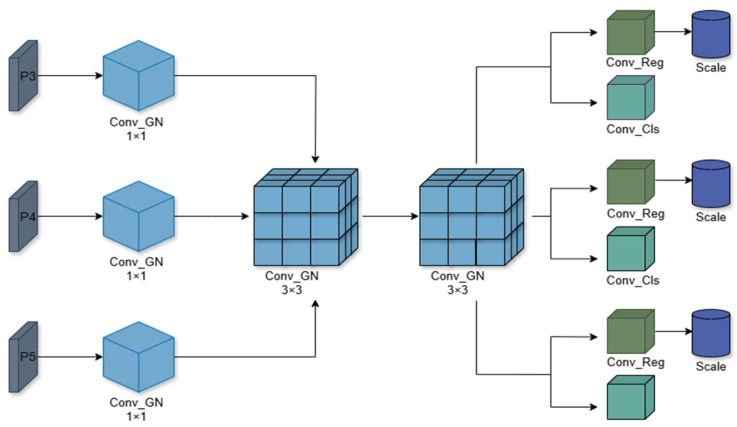
Structure of LSCDH.

**Figure 7 sensors-25-05147-f007:**
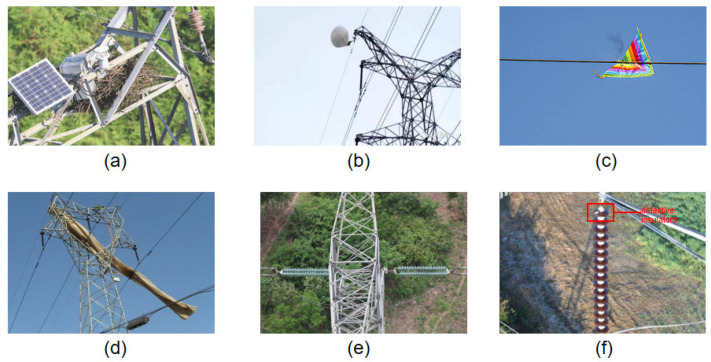
(**a**) bird nests, (**b**) balloons, (**c**) kites, (**d**) debris, (**e**) insulators, (**f**) defective insulators.

**Figure 8 sensors-25-05147-f008:**
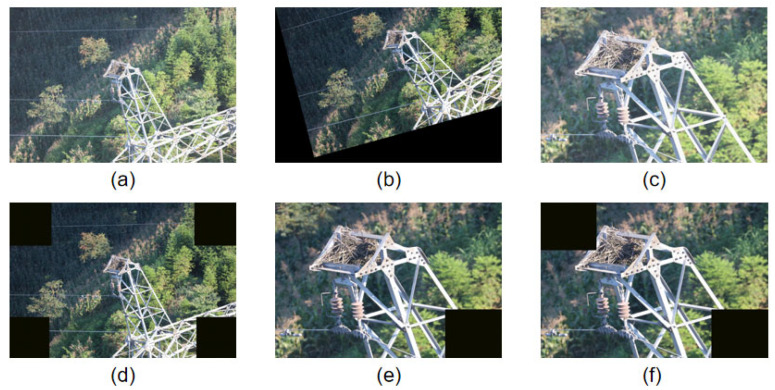
Data augmentation process and results. (**a**) Original image. (**b**) Random rotation. (**c**) Gaussian noise image. (**d**–**f**) Random occlusion.

**Figure 9 sensors-25-05147-f009:**
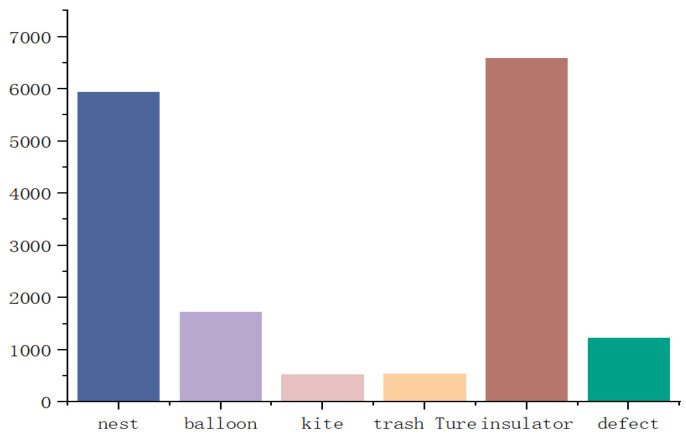
Label classification.

**Figure 10 sensors-25-05147-f010:**
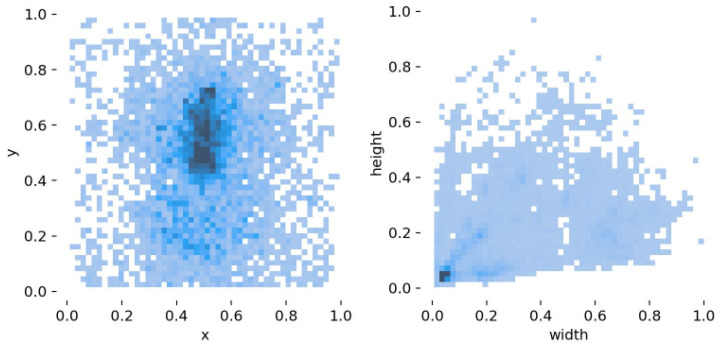
Bounding box distribution maps.

**Figure 11 sensors-25-05147-f011:**
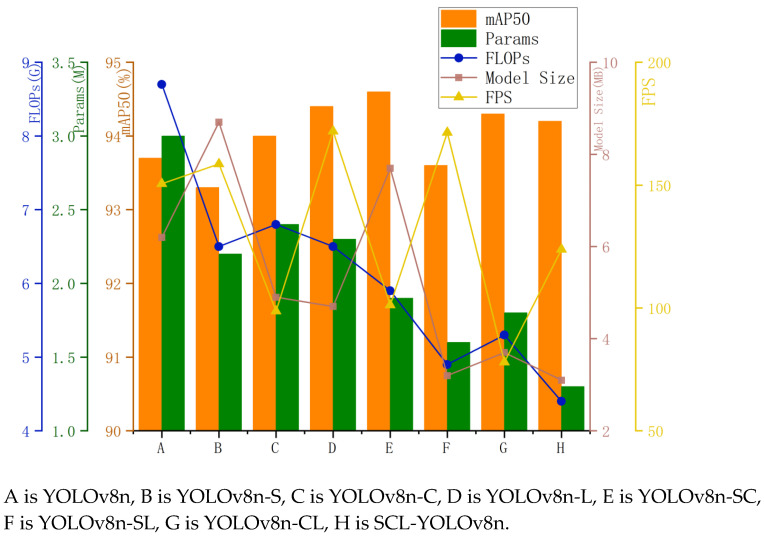
Ablation experiments.

**Figure 12 sensors-25-05147-f012:**
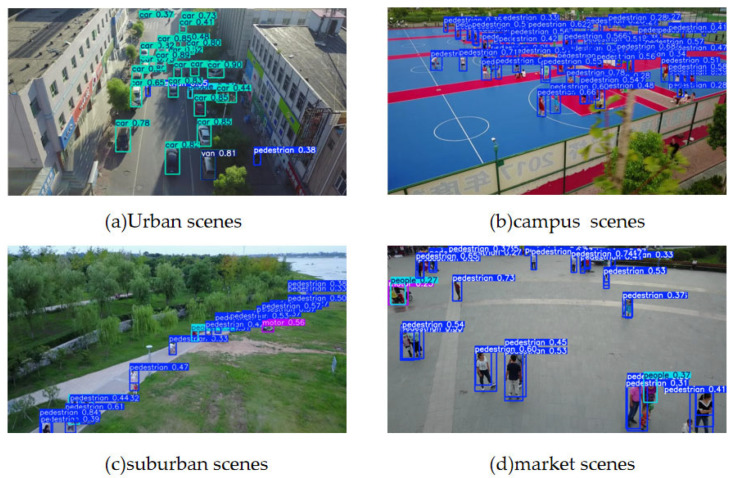
Performance of the VisDrone dataset in SCL-YOLOv8 across different scenarios.

**Figure 13 sensors-25-05147-f013:**
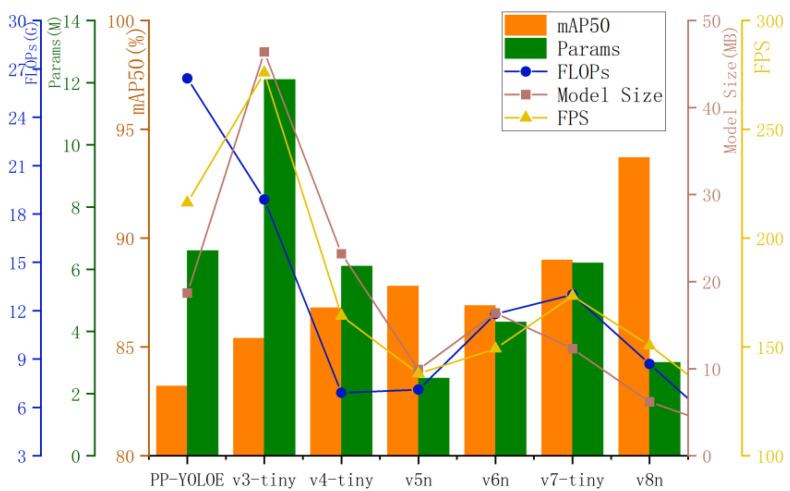
Comparison experiments.

**Figure 14 sensors-25-05147-f014:**
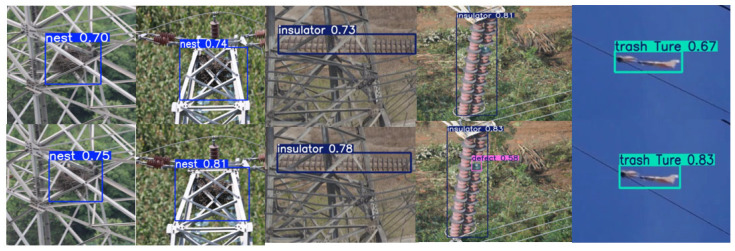
YOLOv8n with added C2f-CGLUCF and LSCDH module visualization comparison plot.

**Figure 15 sensors-25-05147-f015:**
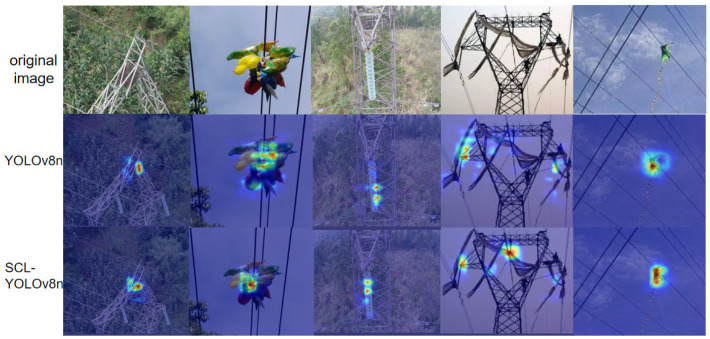
Thermograms of YOLOv8n and SCL-YOLOv8.

**Figure 16 sensors-25-05147-f016:**
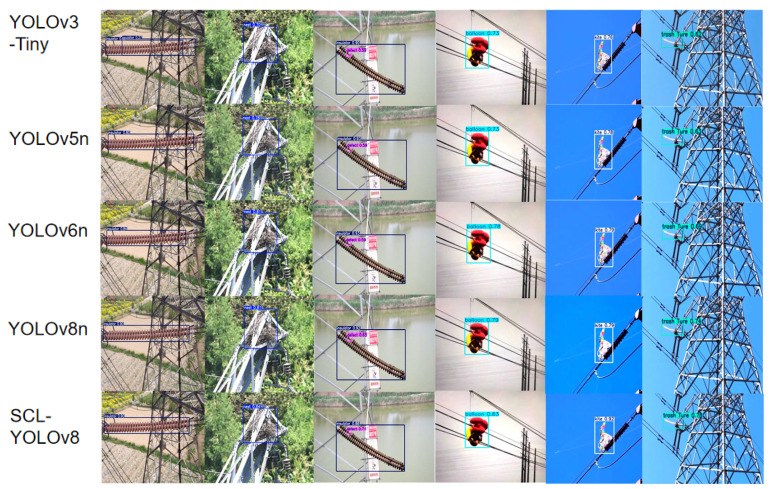
Visualization results of YOLOv3-Tiny, YOLOv5n, YOLOv6n, YOLOv8n and SCL-YOLOv8.

**Table 1 sensors-25-05147-t001:** Model hyperparameter settings.

Parameters	Setup
Image Size	640 × 640
Learning Rate	0.01
Epoch	300
Batch Size	16

**Table 2 sensors-25-05147-t002:** Our dataset compared with RailFOD23.

DataSet	Ours	RailFOD23
Images	9736	14,615
Labels	16,532	49,541
Categories	6	4
Data Source	Real data	Synthetic data
Proportion of Small Targets	35%	20%

**Table 3 sensors-25-05147-t003:** Ablation experiment results.

Model	StarNet	C2f-CGLU-ConvFormer	LSCDH	mAP@ 0.5 (%)	Params (M)	FLOPs (G)	Model Size (MB)	FPS
YOLOv8n				93.7	3.0	8.7	6.2	150.6
YOLOv8n-S	√			93.3	2.2	6.5	8.7	158.6
YOLOv8n-C		√		94.0	2.4	6.8	4.9	98.7
YOLOv8n-L			√	94.4	2.3	6.5	4.7	171.9
YOLOv8n-SC	√	√		94.6	1.9	5.9	7.7	101.2
YOLOv8n-SL	√		√	93.6	1.6	4.9	3.2	171.4
YOLOv8n-CL		√	√	94.3	1.8	5.3	3.7	77.9
SCL-YOLOv8n	√	√	√	94.2	1.3	4.4	3.1	123.8

**Table 4 sensors-25-05147-t004:** Performance of the VisDrone dataset on YOLOv8n and SCL-YOLOv8.

Model	mAP@ 0.5 (%)	Params (M)	FLOPs (G)	Model Size (MB)	FPS
YOLOv8n	42.1	3.0	8.7	16.4	200.3
SCL-YOLOv8	47.8	1.3	4.4	6.0	173.2

**Table 5 sensors-25-05147-t005:** Comparison experiment results.

Model	mAP@ 0.5 (%)	Params (M)	FLOPs (G)	Model Size (MB)	FPS
PP-YOLOE	83.2	6.6	26.4	18.7	216.3
YOLOv3-Tiny	85.4	12.1	18.9	46.4	276.1
YOLOv4-Tiny	86.8	6.1	6.9	23.2	164.3
YOLOv5n	87.8	2.5	7.1	9.9	137.8
YOLOv6n	86.9	4.3	11.8	16.4	149.2
YOLOv7-Tiny	89.0	6.2	13.0	12.3	173.4
YOLOv8n	93.7	3.0	8.7	6.2	150.6
SCL-YOLOv8	94.2	1.3	4.4	3.1	123.8

## Data Availability

The data that support the findings of this study are available from the corresponding author upon reasonable request. The data are not publicly available due to privacy.

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
