# Peer review of "Lightweight SCL-YOLOv8: A High-Performance Model for Transmission Line Foreign Object Detection"

_sensors, 2025, doi:10.3390/s25165147_

Round 1
Reviewer 1 Report
Comments and Suggestions for Authors
This paper proposes a lightweight object detection model, SCL-YOLOv8, based on YOLOv8n, for foreign object detection on transmission lines. The overall framework is well-designed, with clearly defined module innovations. It demonstrates strong practical value in lightweight design and multi-scale feature fusion, making it suitable for engineering deployment. However, the following issues remain:
-
The paper constructs a relatively comprehensive transmission line foreign object dataset (9,736 images across 6 categories), but lacks comparison with existing representative datasets. For instance, RailFOD23 introduces a synthetic dataset, and EPRepSADet performs anomaly detection based on it.
-
Reference citation errors are severe throughout the paper, with multiple instances of “Error! Reference source not found.” These broken references must be corrected.
-
Although some traditional and deep learning-based foreign object detection methods are cited, the connection to the nondestructive testing (NDT) domain—particularly structural health monitoring and fault diagnosis models—is lacking.
-
Example: Keshun Y, Yingkui G, Yanghui L, et al. A novel physical constraint-guided quadratic neural networks for interpretable bearing fault diagnosis under zero-fault sample, Nondestructive Testing and Evaluation, 2025: 1–31.
Additionally, in the context of model compression, the authors should discuss knowledge distillation-based methods to improve the completeness of the literature review. -
Example: S. Yang et al., "Learning From Human Educational Wisdom: A Student-Centered Knowledge Distillation Method," IEEE Transactions on Pattern Analysis and Machine Intelligence, vol. 46, no. 6, pp. 4188–4205, June 2024, doi: 10.1109/TPAMI.2024.3354928.
-
-
As shown in Figure 10, object instances are heavily concentrated near the image center, with a notable distribution of small objects. Although the paper proposes lightweight modules to enhance small object detection, it does not clarify whether any specific mechanisms were designed to address the dataset’s imbalance.
-
The current conclusion spans nearly 30 lines, with repeated mentions of methods, module structures, and comparative results. It is recommended to condense the conclusion to within 150–200 words for clearer logic and more concise expression.
6.While the ablation study is comprehensive, the paper lacks a more detailed explanation of how the three modules (StarNet, CGLU-ConvFormer, LSCDH) interact or complement each other in feature representation. A brief analysis of their synergy would enhance the reader’s understanding of the integrated model design.
Author Response
|
Response to Reviewer 1 Comments
|
||||||||||||||||||||
|
1. Summary |
|
|
||||||||||||||||||
|
We sincerely thank you for taking the time to review our manuscript, sensors-3803423, Lightweight SCL-YOLOv8: A High-Performance Model for Transmission Line Foreign Object Detection, using your professional expertise. We appreciate your constructive comments, which have greatly improved the quality of our work. Below are our detailed responses to all comments, with corresponding revisions marked in blue in the resubmitted file. All page numbers refer to the revised manuscript. |
||||||||||||||||||||
|
2. Questions for General Evaluation |
Reviewer’s Evaluation |
Response and Revisions |
||||||||||||||||||
|
Does the introduction provide sufficient background and include all relevant references? |
Not applicable |
|
||||||||||||||||||
|
Are all the cited references relevant to the research? |
Not applicable |
|
||||||||||||||||||
|
Is the research design appropriate? |
Not applicable |
|
||||||||||||||||||
|
Are the methods adequately described? |
Not applicable |
|
||||||||||||||||||
|
Are the results clearly presented? |
Not applicable |
|
||||||||||||||||||
|
Are the conclusions supported by the results? |
Not applicable |
|
||||||||||||||||||
|
3. Point-by-point response to Comments and Suggestions for Authors |
||||||||||||||||||||
|
Comments 1: The paper constructs a relatively comprehensive transmission line foreign object dataset (9,736 images across 6 categories), but lacks comparison with existing representative datasets. For instance, RailFOD23 introduces a synthetic dataset, and EPRepSADet performs anomaly detection based on it. |
||||||||||||||||||||
|
Response 1: We sincerely appreciate the reviewers' valuable suggestions and would like to express our gratitude for their professional feedback. In response, we have now compared our dataset with RailFOD23 in the dataset section and have highlighted the advantages of our dataset. We also acknowledge that there remains room for further improvement, and in future work, we will focus on refining the dataset further. Regarding the dataset download issue, we encountered a failure due to insufficient permissions. We will make efforts to obtain the complete dataset again in future work. Once successfully acquired, we will conduct generalization experiments to validate the benefits of our proposed improvements. The modified sections in the manuscript are highlighted in red, with specific changes referenced on pages 12 and 13 of the revised draft. The modifications are as follows: The RailFOD23 dataset is a synthetic collection specifically designed to detect foreign objects on railway power lines. It is based on approximately 400 real images and synthesized using Adobe Photoshop (PS) software to create 412 abnormal power line images, alongside automatically generated images via ChatGPT, AIGC, and data based on Railsem19. The dataset consists of 14,615 images with annotations for four types of foreign objects: plastic bags, floating objects, bird nests, and balloons, amounting to a total of 40,541 labels. As shown in Table 2, compared to the transmission line dataset presented in this paper, the RailFOD23 dataset includes more images. Still, it lacks annotations for defective insulators and has a lower proportion of small targets.
Table 2. Ours dataset is compared with RailFOD23.
|
||||||||||||||||||||
|
Comments 2: Reference citation errors are severe throughout the paper, with multiple instances of “Error! Reference source not found.” These broken references must be corrected. |
||||||||||||||||||||
|
Response 2: We sincerely apologize for the errors in the citations in the submitted version. This issue was caused by the final Word field codes not being updated. We have carefully reviewed each reference to ensure it matches the original text and regenerated the citations (using the Zotero tool), strictly adhering to the journal's formatting requirements. Manual review of cross-references. Thank you for pointing out this issue. We will strengthen our final manuscript review process. The specific revisions are as follows: …Once these foreign objects come into contact with conductors, they can easily trigger electrical discharge, tripping, short circuits, or line disconnections, potentially leading to large-scale power outages with significant economic losses and social consequences [2]. (Specific revisions are listed on page 2.) …In the early stages, power departments relied on manual inspection methods for transmission line maintenance [3]. However, these methods exposed workers to significant risks such as falls from height, mechanical injuries, and electrocution, with particularly severe hazards when op-erating in remote mountainous areas [4]. (Specific revisions are listed on page 2.) …However, weather conditions, environmental variability, and limited visibility often affect image quality, making it difficult to ensure consistently high-quality data acquisition [9]. (Specific revisions are listed on page 2.) ConvFormer [35] uses MetaFormer as the base architecture and adopts a 4-stage hierarchical architecture, where features are processed in each stage by downsampling and a different number of modules, which ultimately realizes an efficient image feature extraction and classification process. (Specific revisions are listed on page 7.) …The Gated Linear Unit (GLU) [36] is a channel mixing mechanism composed of two linear projections. (Specific revisions are listed on page 8.) …In LSCDH, a shared convolutional kernel is used to integrate the convolution operation with Group Normalization (GN) [37], forming a unified module termed Conv Group Normalization (Conv-GN). (Specific revisions are listed on page 9) The first part includes 848 images from the publicly available Chinese Power Line Insulator Dataset (CPLID) [38], containing 227 normal and 621 defective insulators, Defective insulators refer to damaged or missing insulators on power transmission lines. (Specific revisions are listed on page 11)
Comments 3: Although some traditional and deep learning-based foreign object detection methods are cited, the connection to the nondestructive testing (NDT) domain—particularly structural health monitoring and fault diagnosis models—is lacking. Example: Keshun Y, Yingkui G, Yanghui L, et al. A novel physical constraint-guided quadratic neural networks for interpretable bearing fault diagnosis under zero-fault sample, Nondestructive Testing and Evaluation, 2025: 1–31. Example: S. Yang et al., "Learning From Human Educational Wisdom: A Student-Centered Knowledge Distillation Method," IEEE Transactions on Pattern Analysis and Machine Intelligence, vol. 46, no. 6, pp. 4188–4205, June 2024, doi: 10.1109/TPAMI.2024.3354928. Response 3: We sincerely appreciate your valuable feedback from a professional perspective. The suggestions regarding strengthening the relevance in the field of non-destructive testing (NDT) and supplementing the knowledge distillation method have enhanced the completeness of our literature review and the quality of our paper. We are deeply grateful for the two relevant references you provided, which have offered us significant assistance and introduced new insights into model compression, thereby pointing us toward new feasible directions for our future work. We have carefully reviewed the relevant literature and added supplementary content on NDT and knowledge distillation methods to the “Related Work” section of the revised draft to enhance our Related Work. Revised sections are on page 4 of the manuscript. The specific revisions are as follows: Structural health monitoring (SHM) and fault diagnosis models in non-destructive testing (NDT) face similar challenges to those in foreign object detection in transmission lines, particularly in real-time anomaly localization and identifying small targets. Keshun et al. proposed a physically constrained quadratic neural network for bearing fault diagnosis under zero-fault sample conditions, where interpretable feature learning is essential for industrial reliability. Similarly, we can adopt their solutions for small target recognition. Fault diagnosis models in NDT have also been applied to CNN and Transformer architectures for foreign object detection. In non-destructive testing (NDT), edge deployment constraints drive the design of lightweight models. Qu et al. achieved real-time insulation angle estimation using deep separable convolutions. However, unlike NDT methods that rely on dedicated sensors, our drone-based approach requires robustness against environmental factors such as lighting and weather changes. In addition to NDT-focused methods that identify small target defects through physical constraints, knowledge distillation (KD) provides an alternative approach for model compression. Yang et al. recently proposed a student-centric knowledge distillation method that dynamically adjusts the learning focus between the teacher and student models. Although KD can reduce model size, our method prioritizes structural optimization over post-training compression, as unmanned aerial vehicle (UAV) edge devices lack the resources to host teacher models during inference, and transmitting line objects (such as deformed kites) requires structural adaptation beyond feature mimicry. Therefore, we prioritize structural optimization, which eliminates the need for pre-training teacher models and maintains end-to-end trainability.
Comments 4: As shown in Figure 10, object instances are heavily concentrated near the image center, with a notable distribution of small objects. Although the paper proposes lightweight modules to enhance small object detection, it does not clarify whether any specific mechanisms were designed to address the dataset's imbalance. Response 2: We sincerely thank the reviewers for their insightful comments on the imbalance issue in our dataset. This suggestion is indeed a key issue in small object detection, making our paper more readable and helping readers better understand the processing we performed on the dataset. To address this issue, we rotated and cropped the dataset, which improved the spatial imbalance of small objects to some extent. The revised content for this section is clearly stated in Section 4.2 Dataset, with specific details on page 12 of the revised manuscript. Additionally, we performed multi-scale feature fusion in LSCDH to enhance sensitivity to small objects. The revised content for this section is clearly stated in Section 3.4 of the LSCDH Module. Specific details are on page 10 of the revised manuscript, with the revised content as follows: 3.4 LSCDH Revision Details The shared convolutional layer preserves both the fine-grained gradients from the high-resolution feature map (P3) and the contextual gradients from the low-resolution map (P5), thereby enhancing sensitivity to small targets. 4.2 Dataset Revision Details: This phenomenon may be attributed to the drone's hovering mode during data collection. To mitigate the data imbalance issue, we employed random rotation, cropping, and scaling techniques to diversify the target locations during dataset preprocessing. In the proposed LSCDH module, multi-scale feature fusion was implemented to enhance sensitivity to small targets.
Comments 5: The current conclusion spans nearly 30 lines, with repeated mentions of methods, module structures, and comparative results. It is recommended to condense the conclusion to within 150–200 words for clearer logic and more concise expression. Response 5: Thank you very much for your valuable feedback! The issues you pointed out are highly professional and to the point. By streamlining repetitive descriptions and module structure explanations, and focusing on core innovations and key conclusions, the academic expression of the paper will become clearer and more impactful. Your suggestions have significantly improved the conciseness and readability of the entire article, which is crucial for enhancing the quality of the paper. We will follow your guidance and condense the conclusion to within 200 words, ensuring that every sentence conveys meaningful information. The specific revisions are outlined on page 18 of the manuscript. Once again, we sincerely appreciate your meticulous and professional review! In this paper, we propose SCL-YOLOv8, a lightweight power line foreign object detection model based on an optimized YOLOv8n framework. This model incorporates three key innovations—StarNet, which replaces CSPDarknet53 with star operations for more efficient feature extraction, CGLU-ConvFormer, which enhances sensitivity to small objects through gated convolutions, and LSCDH, which achieves multi-scale fusion via shared convolutions and group normalization. These advancements allow the model to maintain a mAP@0.5 of 94.2% while simultaneously reducing the number of parameters, FLOPs, and model size by 56.8%, 45.7%, and 50%, respectively. These improvements enable real-time deployment on drones (123.8 FPS) without sacrificing accuracy. Comments 6: While the ablation study is comprehensive, the paper lacks a more detailed explanation of how the three modules (StarNet, CGLU-ConvFormer, LSCDH) interact or complement each other in feature representation. A brief analysis of their synergy would enhance the reader’s understanding of the integrated model design. Response 6: Thank you for your insightful comments! The issues you raised are highly professional and constructive. Indeed, a thorough analysis of the collaborative mechanisms among the three modules—StarNet, CGLU-ConvFormer, and LSCDH—can help readers better understand the overall design logic of the model and the advantages of its feature representation. Your suggestion has made us realize that supplementing the analysis of module interactions not only enhances the theoretical depth of the paper but also highlights the innovative aspects of the integrated model. In the revised version, we have thoroughly explored the complementarity of the three modules. The detailed revisions are outlined on page 15 of the manuscript, with specific changes as follows. Your feedback has significantly improved the coherence and readability of the paper. Once again, thank you for your professional review! The combination of StarNet and CGLU-ConvFormer (referred to as YOLOv8n-SC in Table 3) improves feature extraction through StarNet hierarchical architecture, which extracts rich multi-scale features and implicitly generates high-dimensional representations. These features are then passed to the neck stage of CGLU-ConvFormer, which enhances local detail representation and nonlinear feature transformation using gated convolutions and channel mixing. As a result, the model's accuracy improves, with mAP@0.5 increasing to 94.6%. The LSCDH module, located in the detection head, utilizes shared convolutions and group normalization to decode multi-resolution features more efficiently, while minimizing redundancy. The sequential deployment of these modules ensures a progressive improvement in feature quality from input to output. Compared to the original YOLOv8n, the proposed model achieves 94.2% mAP@0.5, with parameters, FLOPs, and model size reduced to 56.8%, 45.7%, and 50% of the baseline, respectively. These results from ablation studies confirm the benefit of multi-module collaboration in the proposed architecture. The enhanced model demonstrates a more accurate focus on target regions, especially for small and occluded objects. The combined effects of Enhanced Hierarchical Extraction (StarNet), Local Refinement (CGLU-ConvFormer), and Multi-Scale Decoding (LSCDH) enable the network to better distinguish true targets from background noise. The proposed model not only ensures high detection accuracy for foreign objects on power lines but also exhibits exceptional lightweight characteristics, making it highly suitable for deployment on resource-constrained edge devices. This allows for real-time performance while reducing model complexity and improving computational efficiency.
|
||||||||||||||||||||
|
4. Response to Comments on the Quality of English Language |
||||||||||||||||||||
|
Thank you for your suggestions. We have made every effort to polish the language in the revised manuscript. |
||||||||||||||||||||

Reviewer 2 Report
Comments and Suggestions for Authors
This manuscript proposes a lightweight object detection algorithm, SCL-YOLOv8, tailored for foreign object detection on transmission lines using UAV imagery. The authors introduce three key modifications to YOLOv8n: a StarNet backbone, a CGLU-ConvFormer neck module, and a lightweight shared convolutional detection head (LSCDH). Experimental results demonstrate improvements in both model accuracy and efficiency, with reductions in parameters, FLOPs, and model size. However, I have following comments to be addressed.
- Novelty clarification. The architectural improvements are effective, but their novelty relative to prior lightweight YOLO variants (e.g., GhostNet, YOLOv8-tiny) is not clearly differentiated. Clarify how CGLU-ConvFormer and LSCDH offer unique advantages over existing modules.
- Generalization evaluation. All experiments are based on a self-collected dataset. To support broader applicability, results on public datasets (e.g., VisDrone, DOTA) would help validate model generalization.
- Dataset transparency. The custom dataset is not publicly accessible. Clarify annotation criteria (especially for "defective insulators") and whether the dataset can be shared or partially released to support reproducibility.
- Writing and formatting issues. The manuscript includes multiple unresolved references (e.g., "Error! Reference source not found.") and inconsistent formatting. These issues should be corrected for clarity and professionalism.
- Comparative scope. The benchmark comparison lacks several recent lightweight detectors such as YOLOv8-tiny or PP-YOLOE. Including these would better position the proposed model within the current state-of-the-art.
The manuscript addresses a practical problem with a well-structured solution. With clearer novelty articulation, improved writing, and additional public benchmarking, this work could make a valuable contribution to lightweight aerial object detection.
Author Response
Seen the attachment.

Reviewer 3 Report
Comments and Suggestions for Authors
- The paper presents an engineering modification of YOLOv8n by combining existing approaches (StarNet, ConvFormer, shared detection head), making the contribution integrative rather than methodologically novel.
- All experiments are conducted on a single self-constructed dataset specific to Chinese transmission line conditions. The lack of evaluation on public datasets limits the generalizability of the results.
- The analysis does not include per-class performance metrics. Given the presence of small and hard-to-detect objects, it would be useful to show how the proposed model handles each object category.
- The manuscript contains multiple formatting errors such as "Error! Reference source not found" and some duplicated references (e.g. 11 and 27). Editorial revision is needed.
- No source code or repository is provided, which hinders reproducibility and independent verification of the reported performance.
Author Response
Seen the attachemnt.

Round 2
Reviewer 1 Report
Comments and Suggestions for Authors
The paper has been revised to a satisfactory level. Can be received in its current form.
Reviewer 3 Report
Comments and Suggestions for Authors
Most of the reviewer’s comments have been addressed, with some only partially resolved; however, the revisions are sufficient for the manuscript to be accepted for publication.